# Towards Managing Uncertain Geo-Information for Drilling Disasters Using Event Tracking Sensitivity Analysis

**DOI:** 10.3390/s23094292

**Published:** 2023-04-26

**Authors:** Siamak Tavakoli, Stefan Poslad, Rudolf Fruhwirth, Martin Winter, Herwig Zeiner

**Affiliations:** 1Computer Science Department, Maharishi International University, Fairfield, IA 52557, USA; 2School of Electronic Engineering and Computer Science, Queen Mary University of London, London E1 4NS, UK; 3Thonhauser Data Engineering GmbH, 8700 Leoben, Austria; 4JOANNEUM RESEARCH Forschungsgesellschaft mbH, 8010 Graz, Austria

**Keywords:** drilling disaster, feature selection, forward selection, sensitivity analysis, random forest classifier, neural networks

## Abstract

In sub-surface drilling rigs, one key critical crisis is unwanted influx into the borehole as a result of increasing the influx rate while drilling deeper into a high-pressure gas formation. Although established risk assessments in drilling rigs provide a high degree of protection, uncertainty arises due to the behavior of the formation being drilled into, which may cause crucial situations at the rig. To overcome such uncertainties, real-time sensor measurements are used to predict, and thus prevent, such crises. In addition, new understandings of the effective events were derived from raw data. In order to avoid the computational overhead of input feature analysis that hinders time-critical prediction, EventTracker sensitivity analysis, an incremental method that can support dimensionality reduction, was applied to real-world data from 1600 features per each of the 4 wells as input and 6 time series per each of the 4 wells as output. The resulting significant input series were then introduced to two classification methods: Random Forest Classifier and Neural Networks. Performance of the EventTracker method was understood correlated with a conventional manual method that incorporated expert knowledge. More importantly, the outcome of a Neural Network Classifier was improved by reducing the number of inputs according to the results of the EventTracker feature selection. Most important of all, the generation of results of the EventTracker method took fractions of milliseconds that left plenty of time before the next bunch of data samples.

## 1. Introduction

### 1.1. Basic Concepts

In sub-surface drilling, critical or crisis events, such as unwanted influx into the borehole, usually arise not abruptly, but have rather a more gradual nature. For instance, the hazard potential of an unwanted influx into the borehole is small, as long as the influx rate affected by the permeability of the formation is low, and as long as the accumulated volume of the influx down the hole is small, and thus both have no further impact. If the influx rate increases while drilling deeper into a high-pressure gas formation, the chance for a crisis to emerge will increase too. Figure 1 demonstrates the importance of crisis prevention among the interactions in drilling disasters [1].

Sensitivity Analysis (SA) allows for linking of normal operational events to the ones that may arise later to disrupt the overall system, i.e., crisis events. It is not always the case that operational events are understood when manually viewing the raw input data. Instead, a better understanding can be gained through analyzing features in the raw data. The next sections will explain the drilling events of interest and the types of features that can be analyzed. Moreover, it will be explained how those events can be linked using event-based Sensitivity Analysis.

### 1.2. Critical Drilling Operations

There are certain critical operations in the process of decision making for drilling disaster management. The schematic diagram in Figure 2 shows these operations in the time order that they may be applied.

The following sub-sections explain each critical operation.

#### 1.2.1. Crisis Detection

The most important type of decision support is the just-in-time detection of an upcoming critical situation [2]. Note that in contrast to crisis prediction, described later, missing the detection of a critical situation leads to inevitable damages at physical entities, injury for humans, and last but not least financial losses.

#### 1.2.2. Crisis Prediction

Early detection of an upcoming crisis may enable a system to avoid the occurrence of a critical event, or at least to limit its physical and financial impacts. It is evident that the prediction of a possible critical situation is uncertain. The longer the prediction periods are (i.e., the timespans between the predictions made and the actual times the critical events would occur) the more uncertain crisis prediction may be.

#### 1.2.3. Counter-Action Support

Besides the detection of already occurring or possible upcoming crises, there is also a need for counter-action support. This means that the system provides guidelines and suggestions for the particular handling of crises to system operators. This can involve the use of simple electronic crisis handling checklists (similar to the practices used by aircraft crew). In addition, more sophisticated methods, e.g., semantic decision tables or even offline trained classifiers based on the experience of a large amount of rig data (aggregating offline knowledge from multiple rigs and boreholes), might be feasible [3].

#### 1.2.4. Crisis Prevention

Crisis prevention is a type of system interaction where, even without any signs of an upcoming crisis occurring, recommendations for “crisis-prevention operations” (such as ream and wash operations) are provided to a system operator. First, an important aspect of this is the provision of feasible parameters for regular operations. For instance, the parameters to guarantee an appropriate pump startup should be proposed by the system so that the operation does not cause an evolution towards a potential critical situation. Second, a justification for the recommended system actions should also be generated. Third, the system should be aware of whether or not an operator has followed the system recommendations. Thus, in the case that the proposed counter-actions are ignored by operators, the system can adapt the recommendations to the new situation or even issue an alarm.

### 1.3. Response Time

Response time is defined as the time to respond to significant events that may otherwise lead to an operational crisis. The required time strongly depends on the detected critical event. If the reaction is set earlier, it gives a higher chance of managing any crisis that develops, although early intervention may also be associated with more uncertainty as to how an event could develop and unfold. Thus, it would be not enough to answer the question for the timing of the response, as it should be the minimum possible. Nevertheless, it can be stated that the overall magnitude of a response time is in the range of seconds to minutes. For example, for a drill kick, the response time strongly depends on the parameters of the subsurface formation, such as permeability, fluid properties, as well as pressure conditions [4]—therefore, an extension in the magnitude of minutes may be required for the response time. Furthermore, often ream and wash operations are applied to prevent a stuck pipe, usually after a stand is drilled and before connecting a new stand to the drill string. Thus, waiting for a decision to recommend or not such an operation makes no sense if the response time is in the magnitude of the duration of that operation. Each decision support function has to meet estimated response time requirements, as shown in Table 1.

During drilling, the driller needs continuous knowledge about the borehole stability and, even more urgently, about any significant borehole instability. The actual state of the borehole is used for any immediate or any eventual urgently required counter-actions or to make any revisions to improve the drilling plan. Its state is typically evaluated using a number of different information sources and is, as far as possible, incorporated into a geological model. Such information sources usually supply real-time data, measurements at regular intervals, and drilling reports [5].

Real-time data typically originate from sensors mounted at the surface, as well as, in some special cases, from down hole; the data are sampled and provided with an interval in the magnitude of seconds. Real-time data are, in general, used also by third-party data providers, such as mud-loggers and directional drillers, to supply an additional set of information in real time. Other measurements, such as mud properties, are provided in intervals in the magnitude of hours, and drilling reports are available on daily basis. Drilling reports contain much more information about the drilling process, wellbore geometry parameters such as diameter and depth, mud properties, wellbore trajectory data, and descriptions of normal and abnormal situations and corresponding counter-actions.

### 1.4. Features

In heuristic model-building, features usually play an important role. They are used as an input to reduce the required complexity of a subsequent (heuristic) problem-solver. Although feature calculation does not extend the information in the input data, it can prepare such information, in many cases, in a way such that the complexity of a given task reduces dramatically. As a rule of thumb, the lower the complexity of a heuristic model, the better is its generalization property, and the more successful it is at processing data that the model has never seen before [6].

In numerous practical cases, features relevant for solving a problem are well known, but in other cases such knowledge must be created from scratch. A common way to create features for such cases is to integrate domain know-how as much as possible and to create numerous features that might be relevant, sometimes based upon gut instinct. Currently, we need to be able to efficiently process large amounts of feature data, and thus we need to reduce the dimension of the generated features. Out of those, sometimes a huge number of features, the relevant ones, can be identified by use of sophisticated methods such as forward selection or backward elimination in combination with principal or independent component analysis [6]. However, these methods are computationally intensive. Therefore, we need to investigate incremental classification methods that could feasibly lend themselves to the problem of dimensionality reduction.

## 2. Related Work

Currently, there is an important trend in using real-time data streams in the oil and gas community [7]. In particular, the research community and the related drilling industry is very active applying new types of data analysis strategies to build decision systems supporting drilling engineers and companies in their daily work.

While the main purpose of such supporting systems is an improvement in efficiency and a decrease in drilling costs, real-time data analysis can also be used for the discovery of critical events such as kick, pump startup, or lost circulation. Standard approaches for analysis use different reasoning strategies to detect deviations in physical models. A typical example is early symptom detection using real-time data [8]. This has been the case in this work, i.e., the deviated models have been applied to the detection of several symptoms (e.g., sliding or rotational friction) under real-world conditions. Another interesting real-time system for decision support for high-cost oil-well drilling operations is presented in [9]. “DrillEdge” uses case-based reasoning techniques to predict problems and give advice to mitigate them.

Although the application of such standard strategies to observations of real-time data streams to predict critical events is clearly feasible, reliable and stable detection and prediction require robust and novel machine learning strategies in order to promote even more reliable and even safer drilling operations. This problem is far from trivial, and to our best knowledge, most of the work using machine learning methods are works in progress for specific events (e.g., stuck pipe) or look at all unusual events. In most cases, the analysis modules or set of analysis modules are only integrated into research prototypes; examples include [10,11] where stuck pipe events are analyzed.

Another general observation of related research is the fact that a single method is not usually sufficient. It is necessary to develop systems that integrate various machine learning methods (with specific advantages and disadvantages) to solve such complex problems. For example, Murilla [12] proposes the use of a hybrid system based upon fuzzy logic and neural networks to predict a stuck pipe. A decision support system can use influence diagrams [13] that seem well-suited for real-time support and for large and complex drilling decisions where there is some uncertainty. They used Bayesian Decision Networks. They came to the conclusion that solutions that take into account techniques to determine how strongly the result of an influence diagram computation depends on the values of the observable variables (sensitivity analysis) are very promising.

Recently, it also turned out that efficient machine learning strategies allow for the increasing of information content via multiple real-time data stream analysis. Thus, the data cannot only be analyzed on a single rig, but also in real-time operating centers (RTOC), which has several advantages, as discussed by Booth in [14].

Besides the use of simple data-streams, it is also feasible to construct deduced (combined) features in order to enrich the variety of information available for use. Sometimes it is easier for machine learning techniques to work with several “weak” features and combine them into a more powerful classifier (ensemble methods and feature selection). Hence, decision makers tend to spend extra effort on constructing new variables (or features) based upon the original variables in order to help make sense of the initial (or raw) measurements [15,16,17,18]. One important solution to address the problem of scalable data processing in real-time is dimensionality reduction [19,20,21,22,23]. These solutions propose the use of regression-based techniques including Projection Pursuit Regression and Principal Component Analysis for variable transformation. The former technique is one of the most common multivariate regression techniques, while the latter is a special case of the former. Another group of methods, known as cluster analysis, aim to lower the number of data entities by replacing a group of similar ones with a representative data entity [24,25].

Ref. [26] reports that by growing the dimension of input variables, the number of possible regression structures increases faster than exponentially. This issue contributes to the high unreliability of regression methods. In cluster analysis, grouping, which according to [24] is related to the end goal of the user, requires analysis of similarity among the input entities, which are in many cases a time series. This makes cluster analysis a computationally heavy type of process for time-critical applications [24].

Sensitivity analysis has been discussed by [27,28,29,30,31] as a technique to minimize the computational overhead by eliminating the input variables that have the least impact on the system. The majority of sensitivity analysis methods tend to demonstrate the impact of change in one variable on the other by means of a mathematical equation that describes the relationship between them. Methods such as differential analysis [32], Green’s function [33], and coupled/decoupled direct [34] are classified among the analytical sensitivity analysis methods by [32]. However, the non-linear and non-monotonic relationships between inputs and outputs of a given system may not necessarily lend themselves to the use of such analytical methods [35]. Refs. [35,36] tackled the issue of the computational cost of a “double-loop sample-generation strategy,” and the use of restrictive conditions for the evaluation of dependent variables based on independent variables in sampling-based SA methods by proposing an approximation approach that measures the entropy of variable distributions in the original data samples. However, obtaining the appropriate indicator functions for each independent variable requires knowledge of their distribution probabilities [36]. EventTracker Sensitivity Analysis supports a reduction of computational cost by eliminating the less-significant features in a timely manner and without use or building knowledge of the distribution of the data.

## 3. EventTracker Sensitivity Analysis

The idea behind the proposed EventTracker platform is the assumption that modern information management systems are able to capture data in real time and have the technological flexibility to adjust their services to work with specific sources of data/information. However, to enable this adaptation to occur effectively in real time, online data needs to be collected, interpreted, and translated into corrective actions in a concise and timely manner. This cannot be handled by existing sensitivity analysis methods because they rely on historical data and lazy processing algorithms. It is important to note that real-time data acquisition and collection systems are equipped with data exchange middleware that have some limited and controlled caching and queuing mechanism such that the published data from the sources of data are not lost until they are collected by the consumer of the data, e.g., the EventTracker platform, in the case of recovery from a malfunction in network connectivity.

In event-based systems, the effect of system inputs on a state is of value, as events could cause this state to change. This “event-triggering” situation underpins the logic of the proposed approach. The event-tracking sensitivity analysis method describes the system variables and states as a collection of events.

An event-based sensitivity analysis method (EventTracker) was proposed by [37]. The proposed event-tracking SA method uses an input-output occurrence [+, −] matrix. This matrix is populated at predefined time intervals. The current platform is designed to allow a user (with domain knowledge) to set the initial system update time interval. For example, in safety sensitive systems such as power plant reactor monitoring, the rate of data table population will be in short intervals. In scenarios that employ less time-critical systems, such as finance, the interval will be longer. This matrix is designed to map the relationships between causes that trigger events (Trigger Data or TD) and the data that describe the actual events (Event Data or ED). In this way, the “EventTracker” method is able to construct a discrete event framework [37] where events are loosely coupled with respect to their triggers for the purpose of sensitivity analysis.

The algorithm is designed to respond quickly, and in essence has a life cycle that is equivalent to a Search Slot (SS). Within each SS, TDs and EDs are captured from two time series and used to provide a value which is translated into a sensitivity index. This index is then added to the indices of the subsequent search slots. At the end of each SS, the sensitivity indices of all data series are linearly normalized. The main functions of the EventTracker algorithm are depicted in Figure 3 and Figure 4. The main steps of the algorithm are as follows.

### 3.1. Stepwise Scan

A First-In-First-Out queue is allocated to every data batch in an SS. The size of the queues is unbounded. The content of the queues is flushed at the end of each SS. The data is then passed to the EventTracker detection and scoring algorithm. The next SS continues to fill the queue immediately. Using this technique, no data is lost. Figure 3 shows a few stepwise scans and their analysis operations in the search slots.

### 3.2. Trigger–Event Detection

Figure 4 shows that within each SS a pair of {ED, TD} are examined for evidence of a trigger and event. The batch of TD values is searched for fluctuations greater than the specified TT threshold, and ED values are similarly checked for changes larger than the ET threshold. This functionality results in a true value being generated, provided at least one of the above changes is found in a particular batch.

### 3.3. Give-and-Take Matching Score

In each SS, the simultaneous existence or non-existence of a change in each pair of data batches is scored as +1, otherwise the score is −1. This operation is similar to a weighted logical Exclusive-NOR and is shown in Table 2. This aggressive approach is adopted to better emphasize the impact of inputs on a given output rather than simply scoring +1 for existence and 0 for non-existence.

### 3.4. Summation of the Matching Scores

The +1 and −1 score for each SS is added to the overall score depicted by Equation (1). The Sensitivity Index (*SI*) of the measured ED and TD values after time t (or in discrete form after search slot), where *n* is the number of SS in an AS, can be calculated as:
(1)$$SI_{\left(t\right)}=\overset{n}{\underset{1}{\sum }}Search\mathrm{Slot}Scores$$

### 3.5. The Normalization Process

At the end of each SS, the values of the sensitivity indices are normalized (2). In other words, given a lower bound *l* and an upper bound *u* for the set of all indices, each final value of the sensitivity index is transformed to a value in the range [0, 1]; thus:(2)$$\tilde{S}=\frac{SI-l}{u-l}$$

A summary example of the algorithm performance is shown in Table 3. In this table, the flow of matching scores and sensitivity indices (SI1, SI2, and SI3) for one ED with respect to three TDs (TD1, TD2, and TD3) over 10 SS is shown.

Star symbols in Table 3 indicate a detected event or trigger for the values of ED, TD1, TD2, and TD3 within each search slot. Each value of S1, S2, and S3 is −1 or +1 depending on the exclusive match between ED and TD1 and TD2 and TD3, respectively. SIn1 to SIn3 represent Normalized Sensitivity Indices values for SI1 to SI3.

The normalized sensitivity indices (SIn) in Table 3 show that ED is most sensitive to TD3 and least sensitive to TD1. Figure 5 shows the values of SIn.

The overall average SIn values are shown in Figure 6; this figure illustrates the lateral movement of the respective values towards a value that is analogous to a steady state.

### 3.6. Time Efficiency

In order to maintain the analysis in real time, generation of one set of Sensitivity Indices of the data belonging to one Search Slot (SS) is allowed to be as long as one SS. Iteration over all Trigger Data (TD) and Event Data (ED) time series for one SS would require a time period as long as that given by Equation (3).
(3)$$T_{Score_{all}}=T_{Score_{i}}N_{Trigger}N_{Event}$$
in which $T_{Score_{i}}$ is the time required to generate each single Sensitivity Index of one ED with regards to one TD, $N_{Trigger}$ is the number of TDs, and $N_{Event}$ is the number of EDs. Therefore:(4)$$T_{Score_{all}}\leq T_{SS}$$
is the time constraint for real-time analysis of the ETSA. This implies that:(5)$$T_{Score_{i}\left(Max\right)}=\frac{T_{SS}}{N_{Trigger}N_{Event}}$$
if the time taken to generate each single Sensitivity Index of one ED with regards to one TD is shorter than this value, i.e., $T_{Score_{i}\left(Max\right)}$ then the EventTracker SA can follow the data series in real time. This will also be discussed at the end of the following Section 4.

## 4. Experiments and Results

### 4.1. TRIDEC Drilling Support Components

The TRIDEC drilling support system (TDS) monitors the drilling operations performed at drilling rigs. It is scheduled to detect and show trends for critical situations in real time [1]. The system is designed to be used both onsite and offsite at a rig at a real-time operating center (RTOC); see Figure 7. It also guides the drillers during routine operations and presents counter-actions and recommendations for abnormal situations. The RTOC is dedicated to be used by a special stakeholder, the so-called RTOC engineer. That RTOC engineer monitors multiple rigs at the same time and is therefore typically located at a remote operating center.

Such an operation center is connected to multiple rigs. The RTOC engineer is provided with an overview of all monitored rigs, enabling the user to get a clear view of the overall situation. It is possible to switch to a more detailed view in order to monitor a single rig on demand. With this system, the RTOC engineer analyzes long-running trends, giving learning feedback to the system and responding to any provided counter-actions and recommendations.

The following subsections briefly introduce three of the drilling support components relevant to the scope of this paper.

#### 4.1.1. System Training Component

In a self-learning computer system, a training component is of utmost importance. Basically, system training is the process of learning. This component enables users to replay historical rig data, to annotate undetected events, and to provide feedback for any existing events detected. Based on such feedback, the system can build up the learning models for subsequent use in real-world installations.

#### 4.1.2. Data Analysis Component

The purpose of the data analysis component is to provide and prepare information and know-how on historical items such as data, inferences, and relations between them. It features the analysis of these items in a fashion similar to conventional online analytic processing techniques. Unlike such systems, it additionally provides analysis methods for generating new knowledge.

#### 4.1.3. Knowledge Editor Component

The knowledge editor is a component that enables a user to create, update, and delete detected events as well as any proposed counter-actions and recommendations. Moreover, the component provides comfortable views on certain critical events in a time-oriented fashion. In certain cases, particular know-how of human experts is required to directly edit model parameters such as thresholds, weights, data type prototypes, or cluster centers.

Data transmission is a big challenge because of the amount of data with regards to real-time transmission. For data transmission, a standard encoding such as WITSML [38] is used. Such standards are based on XML. For instance, one rig currently provides more than 700 real-time channels sampled with a frequency of 1 Hertz. Figure 8 shows a portion of such data together with the associated information on some states of the drilling rig. The transmission of such a large amount of data from one rig is accomplishable by the use of DSL or satellite communication channels. Continuously receiving such data streams concurrently from numerous rigs may exceed the capacity of such channels.

### 4.2. Feature Construction

Some real-time data channels are shown in Figure 8. Taking, for instance, the stuck pipe scenario into account, it can be presumed a priori that the block position and hook load play some role. If the pipe is stuck, the block cannot be moved with the drill string connected. Attempting to move the block up will drastically increase the hook load without giving the block a chance to move—or, to be more precise, the block speed is nearly zero. So, the block speed, which is the first derivative with regards to time of the block position, might be a feature that simplifies the problem of stuck pipe detection. The same applies to drill string rotation and torque.

In terms of real-world tasks, stuck pipe detection is an actual but rather simple problem. A more complex task is stuck pipe prediction, and, as a consequence, stuck pipe prevention through using some precautionary counter-actions. The assumption that hook load and torque contain information about a possibly emerging stuck pipe is still valid. How that information is provided is actually unknown, and therefore a challenge whose importance should not be underestimated.

Figure 9 sketches a typical borehole drilled nearly vertical at its beginning which then changes direction to nearly horizontal. The main components of the hook load $F_{Hook}$ are the acceleration force $F_{A}$, the component of the total weight of the whole drill string ($F_{T}$) aligned with the drilling direction $F_{W}$, (according to the principles of mechanics the component that is vertical to the drilling direction $F_{N}$ is to be ignored), the friction forces $F_{F}$ (that according to the principles of mechanics could depend on $F_{N}$), and some other non-quantifiable forces denoted as ε. In case of creating a deterministic model, the mass influx of the drill string, the borehole trajectory, especially the inclination and friction factors, amongst others, need to be known.

Since it appears unpromising to identify and estimate all input factors with a reasonable certainty and accuracy to predict and thus prevent a stuck pipe (as well as other crises), a heuristic approach incorporating deterministic know-how seems to be the most feasible solution. To incorporate as much deterministic know-how as possible, a systematic approach to generate features based on specific laws of physics appears to be appropriate.

For feature creation, the physical rules of kinematics and dynamics were applied to the drill string in a first approach to create some heuristics for the features created via the available data channels. For instance, the simple rule for the acceleration force $F_{A}$ based on the mass of *m* and the acceleration of *a*,
(6)$$F_{A}=m\cdot a,$$
leads to the resulting heuristic that $F_{A}$ can be expressed as
(7)$$F_{A}=c_{1}\left(d_{\dot{h}}a+d_{s}a\right)+c_{2}a.$$

In Equation (7), $d_{h}$ denotes the length of the drill string above a rotary table and thus out of the borehole, $d_{s}$ is the length of the drill string in the borehole, and a is the acceleration applied to the total drill string. The constants $c_{1}$ and $c_{2}$ need to be evaluated. If a deterministic model builder is used, they are probably automatically estimated using a heuristic model builder and thus can be ignored if the same features are derived. In fact, the features from Equation (7) are $d_{h}*a$, $d_{s}*a$ and the acceleration a of the drill string itself.

Combining all the kinematic and dynamic rules heuristically and normalizing the results by the length of the drill string, a set of exactly 100 feature rules was obtained. Those rules were applied to the 10 base channels as shown in Table 4 as well as to the extended base channels shown in Table 5, resulting in a total of 1600 features.

The denotation of the features is based on the channel from where it originates (e.g., C0108) and the feature index (1 out of 100, H00 denotes the first and H99 the last of the features). Thus, the denotation C0108:H01 stands for the feature with index-1 based on bit depth, which is in fact the drill string acceleration.

Figure 10 shows a portion of such features generated within a 42-h window. The base channels in that case were sampled with a sampling frequency of 0.1 Hz equivalent to a sampling interval of 10 s. In all charts, the bit depth is assigned to the right axis and drawn as a black line. In Figure 10a, the bit velocity is drawn as a yellow line within a range of about ±0.8 m/s. The bulk of the positive bit velocities on the left (3:20 to 6:10) is due to the trip-in of the drill string into the borehole. The middle part of the chart indicates the actual drilling operation starting at 6:10 and ending the next day at 9:20. Over this period, the average bit velocity is about 16 m/h (equivalent to 5 mm/s) and thus not really perceptible in the chart. The large spikes in that range (e.g., at 14:00) indicate so-called ream and wash operations applied for cleaning the borehole. The right part of the chart shows, again, a period of large bit velocities due to a trip-out operation when the drill string is removed from the borehole.

In Figure 10b, the bit acceleration is drawn against the time and the chart is similar to that above; a period of large acceleration values occurs during a trip-in and trip-out. Some acceleration spikes during drilling are caused by the ream and wash operations. Figure 10c shows the pump power (E0201, cp. Table 5), the product of the pump pressure (C0121), and flow rate (C0130). It is obvious that the most pump power is required for drilling; for the trip-in and trip-out it is almost zero.

### 4.3. Validation

To show the feasibility of the proposed Sensitivity Analysis method for borehole state classification tasks, we performed several experiments on real-world datasets. First, we used the EventTracker Sensitivity Analysis for a reduction of the number of possible features mentioned in Section 1.4 to a manageable, “important” subset. Later, the obtained reduced feature set (features ranked due to their “importance”) was used with standard methods to solve the borehole operational state classification task.

### 4.4. Validation Dataset

The dataset used for our experiments consists of five anonymized well data recordings (S055, S075, S085, S140, and S240) containing about 1.14 million valid data sets in total (297,033, 277,146, 153,145, 207,217, and 208,696). Besides the 10 base channels directly recorded from the rig and 6 calculated basic features (cp. Table 4 for details), 1600 extended features (see Section 4.2 for details) have been calculated and taken into account. An expert team of drilling engineers has labelled each of the data sets according to one of the ten corresponding borehole operational states (see Table 6), thus providing a reliable ground truth for evaluation.

### 4.5. Obtaining a Feature Ranking from SA Algorithm

As a first experiment, the EventTracker SA algorithm was run on the datasets S055, S075, S085, and S140 individually. We looked for indications of correlation among several SA algorithm runs.

For example, Figure 11 shows the raw importance scores (SI as in Table 3), i.e., all Sensitivity Index values obtained for all features obtained from the channel C0108 (mdBit) and CodeOS labels. The correlation of SA importance values among several boreholes can be easily seen. Similar results have been observed for other channels and their deduced features. This motivates the assumption that sorted SA importance values, or, in other words, importance rankings, can be averaged over several boreholes. Thus, we calculate the mean normalized importance values for each of the 10 base channels and 6 base features and take the 2 most significant channels as “SA-selected” features for all of the following experiments. Table 7 shows the most and second-most important features selected by the SA algorithm for several boreholes.

### 4.6. Experiments with a Random Forest Classifier

Experiments were undertaken to assess how well the resulting Sensitivity Indices correlated with the results of other state-of-the-art classification methods, first the Random Forest Classifier (Section 4.6), and then Neural Networks (Section 4.7). Random Forest Classifiers (RFC) [39,40] are well-known classification methods used in the machine learning community. They have shown better, or at least comparable, performance in comparison to other state-of-the-art classification algorithms, such as SVMs [41] or boosting [42], and show a series of advantages such as efficiency when used on large databases and robustness to missing data.

For this experiment, we used an adapted version of the online algorithm proposed by Saffari et al. [43]. All our experiments were performed on MATLAB (R2011-64bit) running on an Intel Xeon 3.2 GHz—Windows 7 machine (Austria, Graz). Due to the randomized nature of the algorithm, we performed 10-fold cross-validation for the datasets. For each validation run, we used a random selection of 90% of the data for learning, while the remaining part of the data was used for testing. The main performance criterion is the correct classification rate (CCR), which is the ratio between the correctly classified datasets and the number of valid test datasets. Due to the unbalanced nature of classes, individual correct classification rates were calculated to emphasise any effects on smaller classes.

At first, we used the 32 most important features suggested by the EventTracker SA algorithm. The results obtained can be seen in the second column of Table 8.

As one can see, the classification rates vary between about 55% and 70%, which indicates the feasibility of the proposed EventTracker SA method. The worst performance for S055 may be explained both by the complexity of the task (This is indicated by the fact that the best possible correct classification rate is also the worst one for this dataset) and the fact that the SA algorithm did not see the specific well data during the selection procedure.

In order to check the performance of RFC using added knowledge from an expert, an offline experiment incorporating an expert’s knowledge for the selection of channels was performed. In particular, we added the first and second derivatives (H01 and H02), as well as the H12 of the base channels C0108 (mdBit), C0110 (mdHole), and C0112 (posBlock) as additional features and obtained the results shown in the third column of Table 8. The results show a correlation between the performance of the EventTracker SA method and the ones from incorporating an expert’s knowledge.

### 4.7. Experiments with a Neural Network Classifier

In addition to the Random Forest Classifiers, Neural Networks were applied to classify the operational states shown in Table 6. In combination with the well-known Sequential Forward Selection (SFS) [6], an estimate of the channels and features relevant for the classification task was made in order to compare it to the features recommended by the SA method. In addition, the performance of the classifier is of interest for comparison to the Random Forest Classifiers.

Neural Networks in general generate a substantive computational load in computers. In addition, feature selection increases this. To constrain the computation time to some acceptable time, a subset of the data was extracted from each of the four wells.

A drilling process is usually separated into so-called runs. For each such run, a rough description of what happened in the well or at the rig is provided. A drilling run includes all the operations applied during the actual drilling of a well and typically consists of trip-in the drill string, drill the well, and concludes with a drilling trip-out of the drill string (see Figure 8).

For the experiments with the Neural Networks, the data were extracted as shown in Table 9. From each well, one single drilling run was selected, and that data was separated into 3 subsets for learning (60%), validation (20%), and testing (20%).

For the classification, a special network architecture, the improved completely connected perceptron (iCCP) shown in Figure 12a, was used. The design of the network layer, number of hidden layers, and number of neurons in each hidden layer, is one of the major challenges in designing a multi-layer perceptron network. The advantage of the iCCP architecture compared to the multi-layer perceptron is that all neurons in the hidden block are completely connected, and thus the search for the optimal network complexity is straightforward. In oil and gas exploration tasks, this architecture has been applied successfully to simulate drilling hydraulics [44].

For all our experiments identical configurations were used. A total of 10 networks were trained in parallel to prevent them from being trapped in local error minima. Network growth was started from scratch, with no neurons in the hidden block, equivalent to multi-linear regression. Then, the hidden neurons were increased one by one until a maximal number of five hidden neurons were obtained.

The number of inputs to all networks was at the utmost of the channels shown in Table 9, but via application of SFS, the input to the networks was managed with forward selection.

In the first experiment, the extracted data from all four wells were used as input to the model. Using all of the 57 channels as inputs to the model, a correct classification rate of about 93% was obtained, but the outcomes were improved by reducing the number of inputs according to the results of the feature selection.

The results of the feature selection are shown in Figure 13, with 10 out of the 57 channels/features selected (4 of those inputs are recommendations from the ETSA method).

The most important input to the model was the difference between bit depth and borehole depth (D0101). The second most important input is the hook load (C0114), followed by rotary speed of the bit (C0120). The CCRs (learning, validation, and test subsets), using only those three channels as input, were all within a range of about 87%. Taking mud flow rate as a fourth input actually provides no improvement to the classification rates, but there is no other channel which provides better results at that position. Adding the bit velocity improves the CCRs by about 4%, to an average of about 92%. Adding more channels according to the SFS rules raises the CCR up to nearly 95%, which is slightly better than the results obtained using all 57 channels as input.

Figure 14 shows the forward selection results of the first SFS cycle. The data are sorted by the model’s validation error (increasing order). There are three channels/features at the left side of the chart providing correct classification rates above 70%, i.e., the difference between bit depth and borehole depth (D0101). The borehole depth was normalized by the drill string length (C0110:H30) and the bit velocity (C0108:H01). The insert in Figure 14 shows the correct classification rates obtained for all 57 channels/features.

### 4.8. Time Efficiency of Applied Sensitivity Analysis

In terms of the time efficiency of the application of EventTracker SA in this paper, according to Equation (3), using a 100 trigger data series (i.e., $N_{Trigger}=100$), 6 event data series (i.e., $N_{Event}=6$), and a search slot period of 100 s (i.e., SS=100 s), the maximum period allowed for generating 1 series of Sensitivity Indices (i.e., $T_{Score_{i}}$) was 160 ms. Generation of SIs took fractions of milliseconds that left plenty of time before the next bunch of data samples filled one search slot.

## 5. Conclusions

Event tracking sensitivity analysis (ETSA) has been applied to facilitate drilling disaster prediction in which the effect of system feature inputs on its state is continuously scored, as input events could cause this state to change. ETSA can reduce the computational burden of dealing with a large number of datasets that may not all necessarily contribute to building an effective processing model. This was demonstrated in this paper by feeding the introduced feature datasets into the ETSA algorithm. It was then observed that the resulting Sensitivity Indices correlated well with the results of other classification methods used in this paper, including Random Forest Classifiers (RFC) and Neural Networks (NN).

Without SA, only expert knowledge decides which of the 100 features is used as an input to the two state-of-the-art event classifiers, RFC and NN. SA was adopted in order to assess whether or not it can provide an alternative method to feature selection that depends on expert knowledge alone or provides a cross-check for it. In the event that expert knowledge is not available, it was shown that the SA outputs could feasibly be used in place of the expert’s input. In addition, the SA (EventTracker) does this in real time. For example, the time performance of the ETSA method was shown to support a time-critical application of data dimensionality reduction for a drilling disaster prediction system. No other SA approach can currently do this in real time.

Oil firms would benefit from this research by replacing expensive and time-consuming expert knowledge that is utilized at the stage of feature selection by the introduced EventTracker Sensitivity Analysis method.

In this work, the methods were applied to the data volumes on the scale of 1000 features. Further research could benefit from covering this limitation by applying the same methodology to much a higher scale of data sources and features that apply to the other sectors of industry, such as manufacturing that usually deals with more than 10,000 data sources and features. Such research is well underway by the main author of this article using state-of-the-art computation platforms.

In addition, future work on the development of the ETSA method is expected to be able to generate higher values for the correct classification rate (CCR) so that, as a result, incorporation of expert knowledge is less critical to preventing environment crisis events. There are two ways to achieve this improvement. First, the search parameters of the implemented algorithm for the ETSA method that are currently fixed values will be evaluated adaptively and dynamically. This will allow the events in the data series to be detected more effectively without the use of expert knowledge. Second, the current method measures the sensitivity of the system output with respect to individual inputs. Accounting for this, the impact of the combination of the inputs on the system’s output will be investigated. Although this will result in a more exhaustive algorithm in terms of time and complexity, both Neural Network-based and Random Forest Classifier-based classification methods can benefit from the generated combined sensitivity indices as the number of dimensions to be processed is reduced. This is especially true for the Neural Network-based method, but also for the Random Forest Classifier-based method where a lower number of features might reduce the probability of incorrect decisions in early levels of the trees.

A further interesting aspect for future work would be the development of an ETSA-focused classification algorithm and working to determine the optimal benefit that the features selected by ETSA provide for different classification methods.

## Figures and Tables

**Figure 1 sensors-23-04292-f001:**
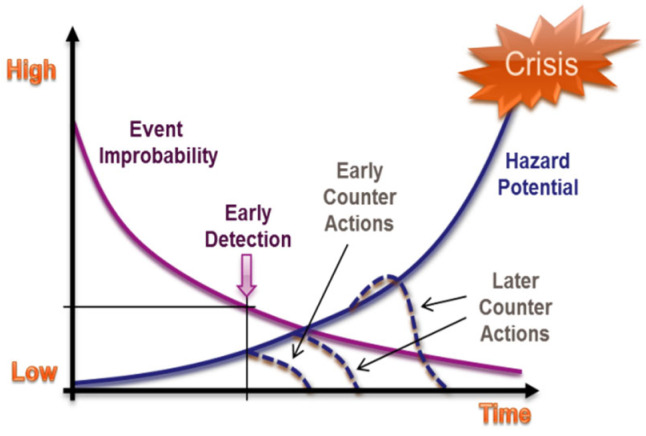
Timeline for drilling interaction leading to a disaster. The vertical axis represents both curves of Event Improbability (the declining curve) and Hazard Potential (the growing curve) ranging between two generally defined bands of low and high.

**Figure 2 sensors-23-04292-f002:**
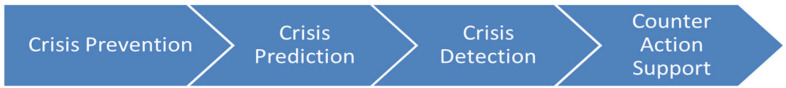
Four Critical Operations in Decision Making for Drilling Disaster Management.

**Figure 3 sensors-23-04292-f003:**
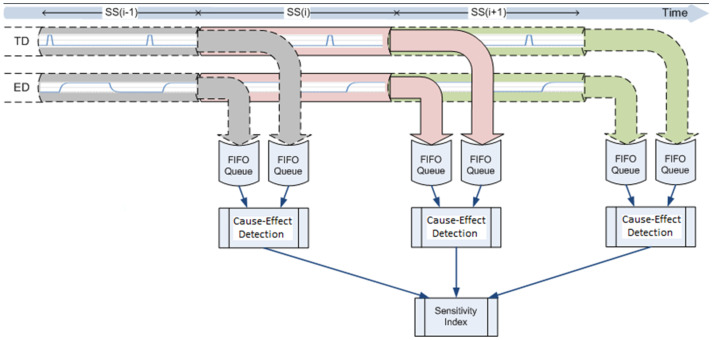
Overall functionality of the EventTracker algorithm.

**Figure 4 sensors-23-04292-f004:**
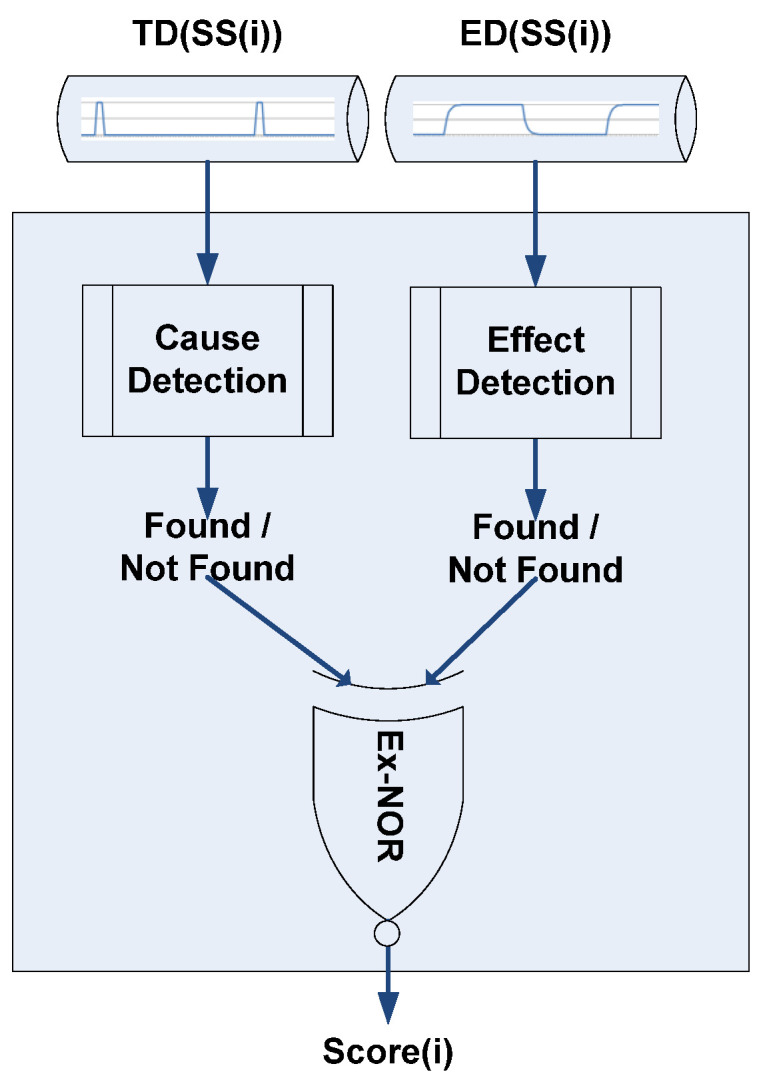
Trigger–Event Detection functionality on each Search Slot.

**Figure 5 sensors-23-04292-f005:**
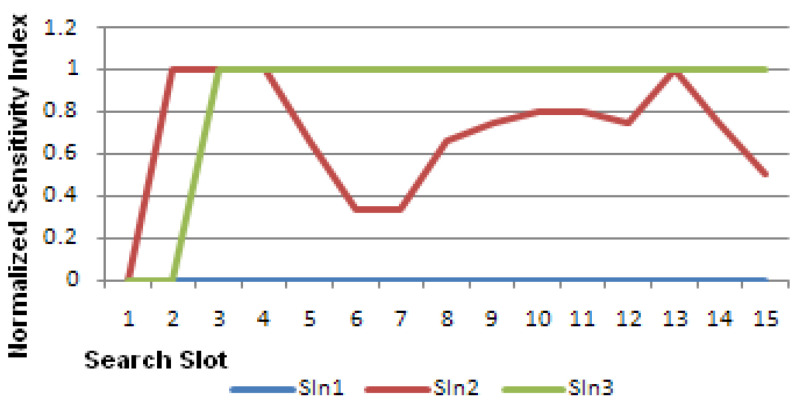
Normalized sensitivity indices as in Table 3.

**Figure 6 sensors-23-04292-f006:**
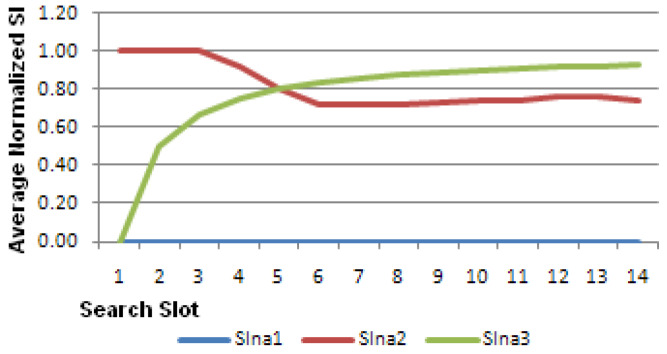
Averaged normalized sensitivity indices for the data in Table 3.

**Figure 7 sensors-23-04292-f007:**
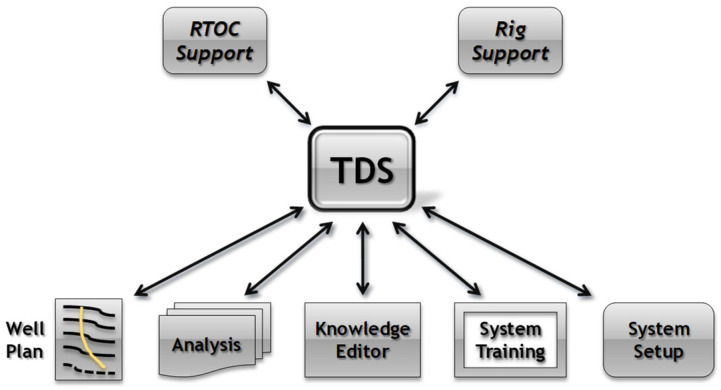
TRIDEC Drilling Support Components (DSC). The different color line in the left block is the plan of the drill in the layers of the soil.

**Figure 8 sensors-23-04292-f008:**
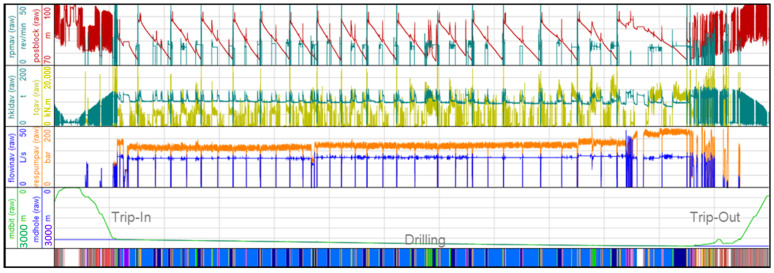
Sensor data time series (four **upper** charts) and state of the drilling (**bottom** chart) for one rig sampled at 0.1 Hz. The top chart shows the block position (red) and drill string rotation speed (green). The second chart shows the torque applied to the drill string (yellow) and the hook load (green). The third chart shows the pump pressure measured at the standpipe (orange) and the mud flow rate (blue). The fourth chart shows the measured depth of the bit (green) and the measured depth of the borehole (blue); also, the main operations trip-in, drilling and trip-out can be identified by the bit depth in this chart. The bottom chart shows the 10 possible operational state labels that may occur at a rig; the predominant light and dark blue encoded states indicate drilling.

**Figure 9 sensors-23-04292-f009:**
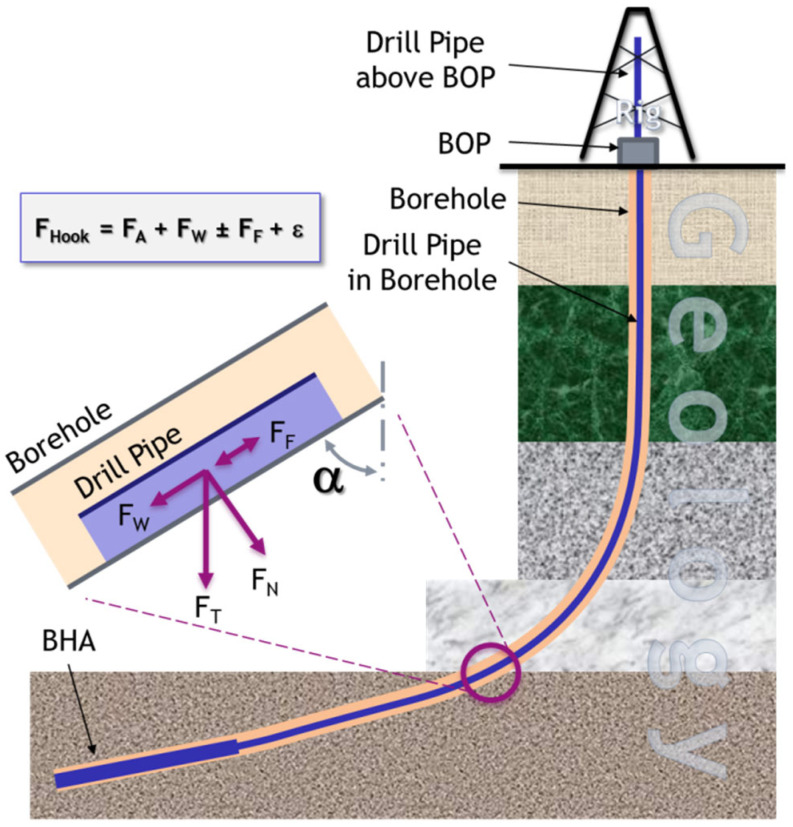
Forces influencing the hook load.

**Figure 10 sensors-23-04292-f010:**
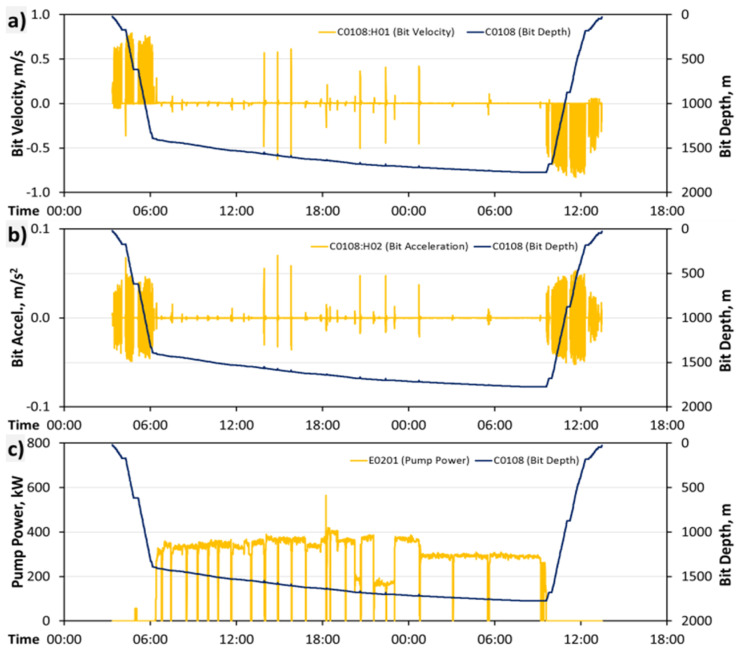
Some of the features based on the sensor data. (**a**) Bit Velocity, (**b**) Bit Accel., (**c**) Pump Power.

**Figure 11 sensors-23-04292-f011:**
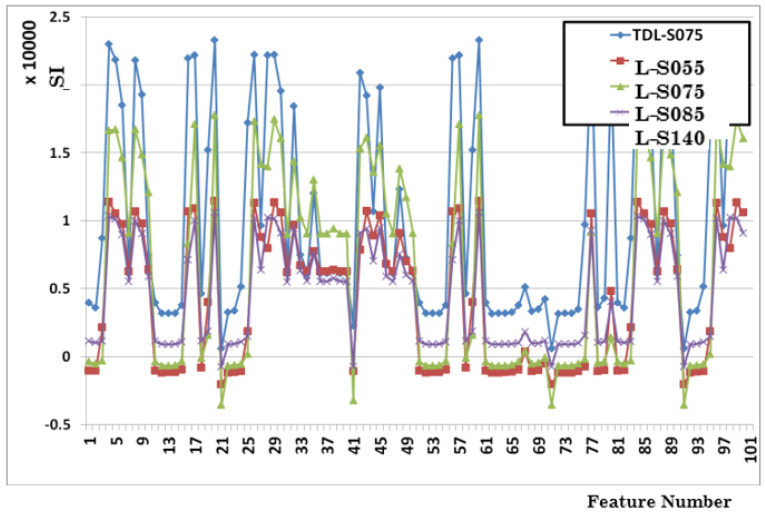
EventTracker SI value correlation behavior.

**Figure 12 sensors-23-04292-f012:**
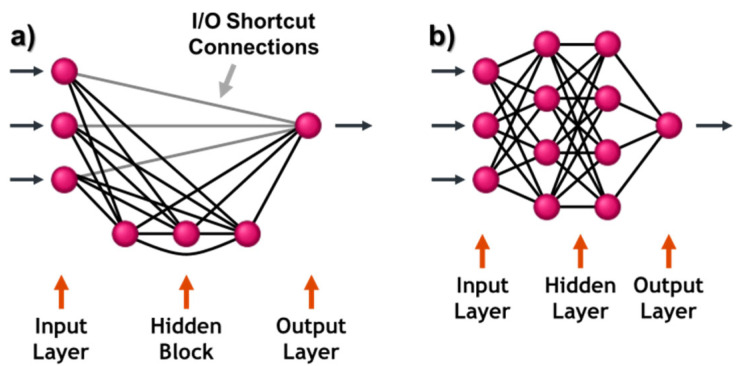
Neural Network Architectures: (**a**) Completely Connected Perceptron and (**b**) Multi-Layer Perceptron.

**Figure 13 sensors-23-04292-f013:**
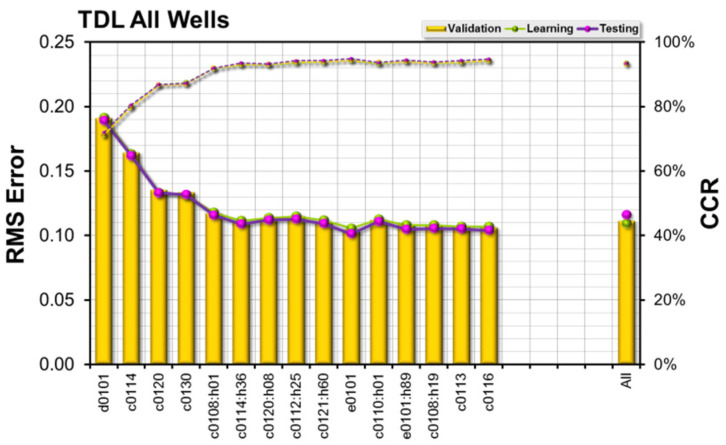
Feature Selection Results—All Wells.

**Figure 14 sensors-23-04292-f014:**
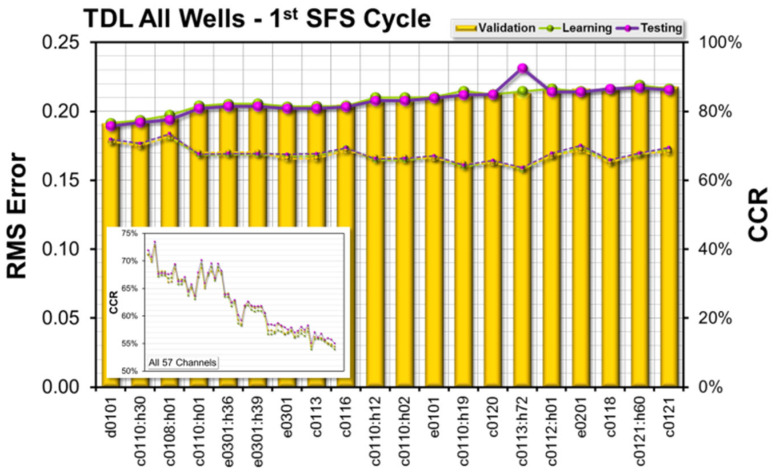
Feature Selection Results—All Wells, 1st SFS Cycle.

**Table 1 sensors-23-04292-t001:** Required response time in drilling disasters.

Event	Response Time
Kick	magnitude of minutes
Stuck pipe support	magnitude of seconds
Pump startup	magnitude of seconds
Lost circulation	magnitude of seconds

**Table 2 sensors-23-04292-t002:** Weighted Exclusive-NOR Functionality.

Input 1	Input 2	Output
0	0	+1
0	1	−1
1	0	−1
1	1	+1

**Table 3 sensors-23-04292-t003:** An example of the sensitivity index generated by the EventTracker method. The star symbol shows occurance of an event.

SS	0	1	2	3	4	5	6	7	8	9	10
ED	*	*		*		*	*	*		*	*
TD1			*			*			*	*	*
S1	−1	−1	−1	−1	1	1	−1	−1	−1	1	1
SI1	−1	−2	−3	−4	−3	−2	−3	−4	−5	−4	−3
SIn1	0.00	0.00	0.00	0.00	0.00	0.00	0.00	0.00	0.00	0.00	0.00
TD2	*				*	*	*	*		*	
S2	1	−1	1	−1	−1	1	1	1	1	1	−1
SI2	1	0	1	0	−1	0	1	2	3	4	3
SIn2	1.00	1.00	1.00	0.67	0.33	0.33	0.67	0.75	0.80	0.80	0.75
TD3		*		*		*		*		*	
S3	−1	1	1	1	1	1	−1	1	1	1	−1
SI3	−1	0	1	2	3	4	3	4	5	6	5
SIn3	0.00	1.00	1.00	1.00	1.00	1.00	1.00	1.00	1.00	1.00	1.00

**Table 4 sensors-23-04292-t004:** Base channels.

ID	Symbol	Unit	Description
C0108	mdBit	m	Total (measured) depth of bit
C0110	mdHole	m	Total (measured) depth of hole
C0112	posBlock	m	Block position
C0113	ropAv	m/s	Drill rate
C0114	hkldAv	kg	Hookload, measured at surface
C0116	wobAv	kg	Weight on bit, measured at surface
C0118	tqAv	J	Rotary torque, measured at surface
C0120	rpmAv	rad/s	Rotary speed, measured at surface
C0121	presPumpAv	Pa	Pump pressure, measured at surface
C0130	flowInAv	m3/s	Mud flow into the hole

**Table 5 sensors-23-04292-t005:** Extended base channels.

ID	Symbol	Unit	Description
D0101	n.a.	m	mdHole − mdBit
D0201	n.a.	m	mdHole + posBlock
D0301	n.a.	m	mdBit + posBlock
E0101	n.a.	W	tqAv ∗ rpmAv
E0201	n.a.	W	pressPumpAv ∗ flowInAv
E0301	n.a.	W	ropAv ∗ wobAv

**Table 6 sensors-23-04292-t006:** Possible Operational States Labels.

State Code	Comments
DrlSld	Drilling Sliding
DrlRot	Rotary Drilling
MakeCN	(Dis)Connect a Drill String
CircHL	Mud Circulation in Borehole
MoveUP	Move Up Drill String
MoveDN	Move Down Drill String
WashUP	Move Up/w Circulation
WashDN	Move Down/w Circulation
CleanUP	Move Up/w Circulation & Rotation
CleanDN	Move Down/w Circulation & Rotation

**Table 7 sensors-23-04292-t007:** Data Subsets used for Feature Selection based on Neural Network Classification.

				Hole Depth, m			Bit Depth, m		
Well	Run Description	Duration	Samples	Min	Max	Span	Min	Max	Span
TDL-S055	Run-5, Drilling	34.1 h	12,281	1407.4	1775.8	368.4	16.6	1775.8	1759.1
TDL-S075	Run-2, Drilling	35.5 h	12,792	158.9	601.9	443.0	21.6	601.9	580.3
TDL-S085	Run-3, Drilling	17.7 h	6361	354.2	624.4	270.2	24.0	624.3	600.4
TDL-S140	Run-5, Drilling	36.7 h	13,202	368.5	1229.0	860.5	21.3	1229.0	1207.7

**Table 8 sensors-23-04292-t008:** Main results for features selected only by the SA algorithm and the best results obtained by taking into account the domain expert’s knowledge.

Well ID	CCR	Best Possible CCR
S055	56%	86%
S075	69%	92%
S085	71%	91%
S140	62%	92%

**Table 9 sensors-23-04292-t009:** List of Base Channels and Corresponding Features Used.

Features (Intuitive Selection)				Features (SA Recommendation)			
Base	1st	2nd	3rd	1st	MI	2nd	MI
C0108	:H01	:H02	:H12	:H19	1.000	:H59	1.000
C0110	:H01	:H02	:H12	:H30	1.000	:H19	0.637
C0112	:H01	:H02	:H12	:H16	0.999	:H25	0.980
C0113				:H17	0.996	:H72	0.994
C0114				:H36	1.000	:H39	0.560
C0116				:H36	1.000	:H39	0.546
C0118				:H63	0.708	:H89	0.682
C0120				:H08	0.994	:H07	0.994
C0121				:H10	0.863	:H60	0.863
C0130				:H01	0.997	:H14	0.994
D0101				:H15	0.995	:H03	0.985
D0201				:H16	0.998	:H04	0.988
D0301				:H16	0.999	:H56	0.995
E0101				:H51	0.930	:H89	0.907
E0201				:H36	1.000	:H39	0.288
E0301				:H36	1.000	:H39	0.980

## Data Availability

Not applicable.

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
