# Peer review of "Towards Managing Uncertain Geo-Information for Drilling Disasters Using Event Tracking Sensitivity Analysis"

_sensors, 2023, doi:10.3390/s23094292_

Round 1
Reviewer 1 Report
1). The key critical crises at the rig should be mentioned in the abstract.
2). The key results are missing in the abstract.
3). Random Forests and Neural network should be included in the keywords.
4). What is the nature of the Critical or crisis events as written in section 1.1. Literature to back them up is needed.
5). An abnormal event in a system that may lead to a critical event can be defined as any anomaly that disrupts the normal operational of a system. The authors should make some statements on errors that are random or difficult to control.
6). A schematic diagram is needed to link all the concepts in section 1.2.
7). A link between feature selection and dimension reduction is needed.
8). List some variables in the related work section that differs based on geology and geography.
9). What happened to the event tracker in the event of sabotage or glitch of internet access.
10). The terms of the equations were not described.
11). Figure 8 should be explained using the principles of mechanics.
12). SA Algorithm and other methods should be introduced in the methodology before the result.
13). What are the evaluation metric results of the classifiers?
14).What are the limitations of the paper?
15). How can Oil firms benefit from this research? Implication for practice is needed.
Author Response
Dear Reviewer,
We, authors of the submitted article, thank you for your invaluable comments, that we tried our best to address in the re-submitted article by modification and addition of the statements where appropriate. The improvements, in accordance to your comments, include the following;
1). The key critical crises at the rig should be mentioned in the abstract.
Added to the beginning of Abstract: In sub-surface drilling rigs, one key critical crisis is unwanted influx into the borehole as result of increasing the influx rate while drilling deeper into a high-pressure gas formation.
2). The key results are missing in the abstract.
Added to the end of Abstract: More importantly, the outcome of a neural Network Classifier was improved by reducing the number of inputs according to the results of the EventTracker feature selection. And most im-portant of all, the generation of results of EventTracker method took fractions of milliseconds that left plenty of time before the next bunch of data samples.
3). Random Forests and Neural network should be included in the keywords.
The two keywords are included.
4). What is the nature of the Critical or crisis events as written in section 1.1. Literature to back them up is needed.
Clarified by modification of the beginning of Secion 1.1 to: In sub-surface drilling, critical or crisis events such as unwanted influx into the borehole, usually arise not abruptly but have rather a smoother nature.
5). An abnormal event in a system that may lead to a critical event can be defined as any anomaly that disrupts the normal operational of a system. The authors should make some statements on errors that are random or difficult to control.
This ambiguous statement is removed. The paragraph starting from slightly modified "Sensitivity Analysis (SA) allows linking normal operational events to the ones that may arise later to disrupt the overall system i.e. crisis events." conveys the message in full and smooth enough to take the reader to the next sentence.
6). A schematic diagram is needed to link all the concepts in section 1.2.
The schematic diagram is included and connected to the following sub-sections.
7). A link between feature selection and dimension reduction is needed.
The authors believe that the last paragraph before Section 2 links the two concepts of feature selection and dimension recuction together. It does so by explaining the difficulty of dealing with the large number of feature data, leading to the intention to reduce the number of features (dimension reduction), and then added to the explaination that due to the compautationally intensive nature of some of the methods such as forward selection and backward elimination, incremental methods that can help with the reduction of dimention are sought.
The authors appreciate to receive clarification if a link of other nature is expected at this stage.
8). List some variables in the related work section that differs based on geology and geography.
Authors appreciate that the reviewer as well as the readers, take the introduced variables involved with the crisis at drilling rigs is independednt of the geology and geography.
9). What happened to the event tracker in the event of sabotage or glitch of internet access.
The following explanation was added to the early parts of Section 3 to cover the case. Further explanation of the middleware caching and queuing mechanisms, such as Kafka data stream and the like, would be outside the scope of this article.
"It is important to notice that real-time data acquisition and collection systems are equipped with data exchange middleware, with some limited and controlled caching and queuing mechanism such that the published data from the sources of data are not lost until they are collected by the consumer of the data, e.g. EventTracker platform in the case of recovery from a malfunction in network connectivity."
10). The terms of the equations were not described.
The description of the equation terms are added and improved.
11). Figure 8 should be explained using the principles of mechanics.
This is further explained and clarified in the modified paragraph where Figure 8 is introduced.
12). SA Algorithm and other methods should be introduced in the methodology before the result.
EventTracker Sensitivity Analysis is introduced in details in Section 3. The established methods Neural Network and Random forest classifiers are referred to by citation to the references [2], [5].
13). What are the evaluation metric results of the classifiers?
The main performance criterion is the correct classification rate (CCR) which is the ratio between the correctly classified datasets and the number of valid test datasets. This is included in Section 4.6.
14).What are the limitations of the paper?
The following paragraph is added to the Conclusion, in addition to the other aspects of the Future Work that was suggested.
The paper demonstrated the improvements that the introduced method had on the drilling disaster data. However, the methods were applied to the data volumes in the scale of 1000 features. Some research could benefit from covering this limitation by applying the same methodology to much higher scale of data sources and features that apply to other sectors of industry, such as manufacturing that usually deal with more that 10,000 data sources and features. Such research is well on the way by the main author of this article using state-of-the-art computation platforms.
15). How can Oil firms benefit from this research? Implication for practice is needed.
The following paragraph is added to the middle of the Conclusion, after the main technical benefits and before the future work.
Oil firms would benefit from this research by replacing expensive and time consuming expert knowledge that is utilized at the stage of feature selection by the introduced EventTracker Sensitivity Analysis method.
Reviewer 2 Report
In this article, Event tracking sensitivity analysis (ETSA) has been investigated in the topic of drilling disaster. Two methods, Random Forest Classifier (RFC) and Neural Networks (NN) have been used. The following items can be checked after reading and reviewing this article:
1. The introduction of the article and the review of references are almost complete.
2. A little more should be explained about work innovation.
3. Preparing a graphical abstract is essential.
4. The Materials and Methods section should be rewritten because currently part of it is given in the Results and Discussion section and is not described as well as the methods.
5. Discuss more about the sensitivity analysis and its results given by NN and RFC methods and explain your results.
Author Response
Dear Reviewer,
We, the authors of the submitted paper, apprecaie your invaluable comments and tried our best to address them, by applying changes and improving the quality of the different sections. Several sections and sub-sections benefitted from additional statements that clarified the intended message.
- The introduction of the article and the review of references are
almost complete.
Authors: The comment is highly appreciated.
2. A little more should be explained about work innovation.
Authors:
- We added to the end of Abstract about the innovative method of dimensionality reduction and performance increase by using EventTracker sensitivity analysis method. We added the following: “More importantly, the outcome of a neural Network Classifier was improved by reducing the number of inputs according to the results of the EventTracker feature selection. And most important of all, the generation of results of EventTracker method took fractions of milliseconds that left plenty of time before the next bunch of data samples.”
- Preparing a graphical abstract is essential.
Authors:
- We provided a schematic diagram that connects the critical operations of decision making process for drilling disaster management. The diagram provides an abstract of the ordered different operations that in turn accommodate the applied methods in this work.
- The Materials and Methods section should be rewritten because
currently part of it is given in the Results and Discussion section and
is not described as well as the methods.
Authors:
- We modified for clarifications: at the beginning of Section 1.1 it is restated as the following: In sub-surface drilling, critical or crisis events such as unwanted influx into the borehole, usually arise not abruptly but have rather a smoother nature.
- We also removed the ambiguity of another paragraph; The paragraph starting from slightly modified "Sensitivity Analysis (SA) allows linking normal operational events to the ones that may arise later to disrupt the overall system i.e. crisis events." conveys the message in full and smooth enough to take the reader to the next sentence.
- We also added and improved the description of the equation terms.
- We also further explained and clarified in the modified paragraph where Figure 8 is introduced.
EventTracker Sensitivity Analysis is introduced in details in Section 3. The established methods Neural Network and Random forest classifiers are referred to by citation to the references [2], [5]. - We also added the following paragraph to the Conclusion, in addition to the other aspects of the Future work that was suggested.
“The paper demonstrated the improvements that the introduced method had on the drilling disaster data. However, the methods were applied to the data volumes in the scale of 1000 features. Some research could benefit from covering this limitation by applying the same methodology to much higher scale of data sources and features that apply to other sectors of industry, such as manufacturing that usually deal with more that 10,000 data sources and features. Such research is well on the way by the main author of this article using state-of-the-art computation platforms.”
- We also added the following paragraph to the middle of the Conclusion, after the main technical benefits and before the future work.
“Oil firms would benefit from this research by replacing expensive and time consuming expert knowledge that is utilized at the stage of feature selection by the introduced EventTracker Sensitivity Analysis method.”
- We also added the following explanation to the early parts of Section 3 to cover the case of network communication failure. Further explanation of the middleware caching and queuing mechanisms, such as Kafka data stream and the like, would be outside the scope of this article.
- "It is important to notice that real-time data acquisition and collection systems are equipped with data exchange middleware, with some limited and controlled caching and queuing mechanism such that the published data from the sources of data are not lost until they are collected by the consumer of the data, e.g. EventTracker platform in the case of recovery from a malfunction in network connectivity."
- Discuss more about the sensitivity analysis and its results given by
NN and RFC methods and explain your results
Authors:
- EventTracker Sensitivity Analysis is introduced in details in Section 3. The established methods Neural Network and Random forest classifiers are referred to by citation to the references [2], [5]. The main performance criterion is the correct classification rate (CCR) which is the ratio between the correctly classified datasets and the number of valid test datasets. This is included in Section 4.6.
Reviewer 3 Report
Dear Sirs, you have produced an exceptional work in which the high level of approach is intertwined with the innovative character, which is why I congratulate you. In order to increase the level of operational security, it is necessary to further develop this analysis system that provides a priori knowledge of possible events or technical accidents. Exceptional. Bravo.
Author Response
Dear Reviewer,
We- the authors of the submitted article, appreciate your comments, and are grateful for your positive attitude towards the work.
Round 2
Reviewer 1 Report
No more comments.
Reviewer 2 Report
In my view, the article appears to be satisfactory based on the reviews. However, to arrive at a final decision, it is advisable to consider the opinions of other reviewers as well.